# Innovative Therapeutic Delivery of Metastasis-Associated in Colon Cancer 1-Suppressing miRNA Using High Transmembrane 4 L6 Family Member 5-Targeting Exosomes in Colorectal Cancer Mouse Models

**DOI:** 10.3390/ijms25179232

**Published:** 2024-08-26

**Authors:** Byung-Jo Choi, Dosang Lee, Jung Hyun Park, Tae Ho Hong, Ok-Hee Kim, Sang Chul Lee, Kee-Hwan Kim, Ho Joong Choi, Say-June Kim

**Affiliations:** 1Department of Surgery, Daejeon St. Mary’s Hospital, College of Medicine, The Catholic University of Korea, Daejeon 34943, Republic of Korea; bottlebird@catholic.ac.kr (B.-J.C.); zambo9@catholic.ac.kr (S.C.L.); 2Catholic Central Laboratory of Surgery, College of Medicine, The Catholic University of Korea, Seoul 06591, Republic of Korea; dosangs@catholic.ac.kr (D.L.); angle49@catholic.ac.kr (J.H.P.); gshth@catholic.ac.kr (T.H.H.); ok6201@hanmail.net (O.-H.K.); keehwan@catholic.ac.kr (K.-H.K.); 3Department of Surgery, Seoul St. Mary’s Hospital, College of Medicine, The Catholic University of Korea, 222 Banpo-daero, Seocho-gu, Seoul 06591, Republic of Korea; hopej0126@catholic.ac.kr; 4Department of Surgery, Eunpyeong St. Mary’s Hospital, College of Medicine, The Catholic University of Korea, Seoul 03312, Republic of Korea; 5Translational Research Team, Surginex Co., Ltd., Seoul 06591, Republic of Korea; 6Department of Surgery, Uijeongbu St. Mary’s Hospital, College of Medicine, The Catholic University of Korea, Gyeonggi 11765, Republic of Korea

**Keywords:** colorectal cancer, exosome, MACC1, microRNA, targeted therapy, TM4SF5 protein

## Abstract

Elevated metastasis-associated in colon cancer 1 (MACC1) expression in colorectal cancer patients, and high transmembrane 4 L6 family member 5 (TM4SF5) protein expressed on various solid tumors’ surface, are linked to aggressive cancer behavior and progression. In this study, adipose-derived stem cells (ASCs) were engineered to produce exosomes (Ex) that target the TM4SF5 protein on tumors. Moreover, MACC1-targeting microRNA was encapsulated within the Ex, resulting in TM4SF5-targeting Ex (MACC1-suppressing miRNA; miR-143). The anticancer effects of these Ex were investigated in vitro using the human colorectal cell line HCT116 and in vivo using colorectal cancer mouse xenograft models. In the in vivo assessment, administration of TM4SF5-targeting Ex[miR-143], referred to as tEx[miR-143] herein, resulted in the smallest tumor size, the lowest tumor growth rate, and the lightest excised tumors compared to other treatments (*p* < 0.05). It also led to the decreased expression of MACC-1 and anti-apoptotic markers MCL-1 and Bcl-xL while inducing the highest expression of pro-apoptotic markers BAX and BIM. These results were consistent with in vitro findings, where t Ex[miR-143] demonstrated the highest inhibition of HCT116 cell migration and invasion. These findings highlight the potential of tEx[miR-143] as an effective therapeutic strategy for colorectal cancer, demonstrating promising results in both targetability and anti-tumor effects in vitro and in vivo, warranting further investigation in clinical settings.

## 1. Introduction

In recent years, the development of exosome engineering technology has led to a growing interest in targeted exosomes as a promising approach for delivering therapeutics, particularly for cancer treatment [1]. Exosomes (Ex), as natural nanoscale vesicles secreted by cells, possess unique characteristics that make them ideal for targeted drug delivery [2]. They exhibit biocompatibility, low immunogenicity, and an inherent ability to transfer biomolecules, such as proteins, lipids, and nucleic acids, between cells [3,4,5]. Additionally, they possess the ability to cross biological barriers, making them suitable candidates for delivering various therapeutic agents, including small molecules, siRNAs, and miRNAs, to specific target cells in the body [6].

Transmembrane 4 L6 family member 5 (TM4SF5) protein is highly expressed in various solid tumors, including colorectal cancer (CRC), where it has been reported to be overexpressed in 58% of cases [7]. It has been shown to play a crucial role in cancer progression processes, such as cell survival, proliferation, migration, and invasion [8]. The overexpression of TM4SF5 in tumor cells makes it an attractive target for cancer therapeutics, as it can serve as a molecular marker for the selective delivery of treatment to cancer cells while minimizing off-target effects on healthy cells [9]. Furthermore, targeting TM4SF5 has the potential to disrupt critical signaling pathways involved in tumor growth and metastasis, leading to improved therapeutic outcomes [10].

Metastasis-associated in colon cancer 1 (MACC1) is a protein involved in the pathogenesis of CRC, as it has been associated with aggressive cancer cell behavior, enhanced metastatic potential, and reduced overall survival in patients [11,12]. The upregulation of MACC1 expression has been linked to advanced tumor stages, lymph node metastasis, and distant metastasis, with an incidence rate of 72.9% in CRC patients [13]. Inhibition of MACC1 expression could potentially lead to significant anti-tumor effects by disrupting key molecular pathways involved in tumor growth, angiogenesis, and metastasis [14]. Therefore, the development of a targeted therapeutic approach that effectively suppresses MACC1 expression holds promise for improving the treatment of CRC patients.

In this study, we hypothesize that TM4SF5-targeting Ex (miR-143) can effectively deliver MACC1-suppressing miRNA to CRC cells, resulting in enhanced anti-tumor effects [15]. By combining the targeting capabilities of TM4SF5 with the therapeutic potential of MACC1 inhibition, we aim to develop a novel, targeted therapeutic strategy that holds promise for improving the treatment of CRC patients [16]. This innovative approach leverages the advantages of exosome-based delivery systems, such as biocompatibility and the ability to cross biological barriers, while exploiting the specific targeting of TM4SF5-expressing tumor cells [17]. Our research aims to provide valuable insights into the design and development of targeted exosome-based therapies for CRC, paving the way for further investigation and potential clinical translation of this novel therapeutic strategy [18].

## 2. Results

### 2.1. Anti-Tumor Effects of TM4SF5-Targeting Ex Encapsulating miR-143 in HCT116 Cells

The nomenclature in this paper is as follows: Ex stands for Exosome(s), and tEx refers to TM4SF5-targeted Exosome(s). When miR-143 or miR-940 is encapsulated within exosomes, they are denoted as Ex[miR-143] and Ex[miR-940], respectively. Since miR-143 was primarily encapsulated into Ex or tEx, after the experiments comparing these two substances, Ex[miR] or tEx[miR] indicates Ex[miR-143] or tEx[miR-143]. Firstly, tEx encapsulating MACC1-suppressing miRNA were successfully fabricated through a series of steps (Figure 1A). Briefly, the DNA sequence of the TM4SF5-targeting peptide was inserted into the pDisplay vector containing a portion of the PDGFR transmembrane domain using restriction enzymes. The resulting recombinant vector was employed to transfect adipose-derived stem cells (ASCs). Transfected ASCs expressed the TM4SF5-targeting peptide on their cell membrane, and the released Ex displayed the TM4SF5-targeting peptide on their surface. Finally, MACC1-suppressing miRNA was encapsulated within the Ex to complete the fabrication of tEx[MACC1-suppressing miRNA]. Based on the Zetaview analysis, the nanoparticles demonstrated a size distribution of 148 ± 84 nm, which falls within the characteristic size range for exosomes (Figure 1B top). This observation was further corroborated by the TEM images (Figure 1B bottom).

Initially, the therapeutic effects of Ex and tEx loaded with either miR-940 or miR-143, two MACC1-suppressing miRNAs, were compared in HCT116 cells using western blotting (Figure 1C). Ex and tEx loaded with miR-940 and miR-143 are denoted as Ex[miR-940], Ex[miR-143], tEx[miR-940], and tEx[miR-143], respectively. No significant difference in MACC1 expression inhibition was observed between tEx[miR-940] and tEx[miR-143]. However, tEx[miR-143] demonstrated a greater reduction in the anti-apoptotic markers MCL-1 and Bcl-xL, and an increased expression of the pro-apoptotic marker PUMA, indicating a higher pro-apoptotic effect (*Ps* < 0.05). This trend was also observed in other CRC cell lines, including HT29 and SW480 cells (Appendix A). Subsequently, an MTT assay was performed to evaluate cell viability in the SW480, HT29, and HCT116 cell lines following each treatment. The results indicated that treating cell lines with tEx[miR-143] led to a greater decrease in cell viability compared to Ex[miR-143] (Appendix A). Subsequently, doxorubicin-resistant CRC cells, HCT116-R cells, and TH29-R cells, were developed, and the cell viability of these cells was measured and compared after treatment with Ex[miR-143] and tEx[miR-143] for 24 h and 48 h, respectively. The results indicated that the tEx[miR-143] treatment group showed a significantly greater reduction in viability of HCT116-R cells and TH29-R cells compared to the Ex[miR-143] treatment group (*p* < 0.05) (Appendix A). Consequently, miR-143 was selected as the Ex-encapsulated microRNA for further experiments to investigate the anti-tumor effects of tEx[miR-143].

To confirm the successful expression of the TM4SF5 peptide in tEx, we performed flow cytometric analysis utilizing Ex markers (CD63 and CD81) and the tEx marker (myc) (Figure 1D). The analysis revealed no significant differences in the expression levels of Ex markers CD63 (33.1% vs. 45.6%) and CD81 (59.0% vs. 54.1%) between Ex and tEx groups, respectively. However, tEx exhibited a substantial increase in the expression of the targeted exosome marker myc compared to Ex, as observed in both CD63-positive (33.6% vs. 0.4%) and CD81-positive (25.5% vs. 0.3%) cells, with a statistically significant difference (*p* < 0.05).

### 2.2. In Vitro Targetability and Efficacy of Intravenously Administered tEx[miR]

Real-time PCR was used to compare the expression of MACC-1, MCL-1, and BAX-1 in four groups: Ex, Ex[miR], tEx, and tEx[miR] (Figure 2A). Hereafter, miR refers to miR-143. tEx[miR] showed the most significant reduction in MACC-1 expression, the lowest expression of the anti-apoptotic marker MCL-1, and the highest expression of the pro-apoptotic marker BAX-1 (*Ps* < 0.05).

Next, the effects of each treatment on HCT116 cell migration and invasion were assessed using a wound healing assay (Figure 2B). The cells were treated with control (Ct), Ex, Ex[miR], tEx, and tEx[miR] and monitored over 48 h. The images and quantification revealed that the tEx[miR-143] group exhibited the most significant inhibition of cell migration and invasion compared to the other groups. Specifically, the tEx[miR] treatment resulted in the smallest wound closure, indicating a substantial reduction in cell motility. This was quantified as a percentage of the initial wound area, with the tEx[miR] group showing a significantly lower cell invasion percentage (*p* < 0.05). This trend was also observed in other CRC cell lines, including HT29 and SW480 cells (Appendix A).

Subsequently, immunofluorescence analysis was performed to determine how the expression levels of E-cadherin, Snail, and Vimentin in HCT116 cells were altered by treatment with various Ex formulations (Figure 2C). Compared to the control (Ct) group, the treatment groups—Ex, Ex[miR], tEx, and tEx[miR]—showed significant changes. Specifically, when comparing Ex to Ex[miR] and tEx to tEx[miR], the epithelial marker E-cadherin significantly increased, while the mesenchymal markers Snail and Vimentin significantly decreased (*Ps* < 0.05). Among the comparisons of Ct, Ex, Ex[miR], tEx, and tEx[miR], the tEx[miR] group exhibited the greatest inhibition of epithelial–mesenchymal transition (EMT), suggesting it has the most substantial effect in reducing EMT, a key process in cancer metastasis. This trend was also observed in the immunofluorescence analysis using other CRC cell lines, including HT29 and SW480 cells (Appendix A). In addition, experiments using spheroid cultures were conducted to simulate the tumor microenvironment and assess the impact of exosomal treatments on ECM factors and cell viability, revealing that the tEx[miR] treatment group exhibited a significant reduction in cell viability compared to other groups (Appendix A).

### 2.3. In Vivo Targetability and Efficacy of Intravenously Administered tEx[miR]

The targetability and efficacy of intravenously administered tEx[miR] were investigated in vivo. To achieve this, mouse CRC xenograft models were established using HCT116 cells, and targetability and efficacy tests were conducted independently (Figure 3A). In the targetability test, Ex[miR] was labeled with fluorescence (ExoGlow) and imaged using the IVIS Imaging system. The images, taken at 2 h and 4 h post injection, revealed that tEx[miR] exhibited higher targetability compared to control (Ct) and Ex[miR] groups in all groups (Figure 3B left). Quantification of the average radiant efficiency in the whole body confirmed the significantly higher targetability of tEx[miR] compared to Ex[miR] at both time points (Figure 3B right). Furthermore, the fluorescence was also measured in the tumors excised from the mouse xenograft models. The comparison of radiant efficiency in the excised tumors demonstrated a higher targetability of tEx[miR] compared to Ct and Ex[miR] groups (Figure 3C). Representative images of the excised tumors indicated higher accumulation of tEx[miR] compared to Ex[miR] (Figure 3C left). Quantification of the average radiant efficiency in the tumors further confirmed the enhanced targetability of tEx[miR] (Figure 3C right).

### 2.4. In Vivo Efficacy of tEx[miR] in Mouse CRC Xenograft Model

Efficacy tests were conducted to evaluate the therapeutic potential of tEx[miR] in mouse CRC xenograft models. On day 15 post treatment, the smallest tumor size was observed in the group treated with tEx[miR] (Figure 4A). The changes in tumor size over the treatment period were monitored, revealing that the tEx[miR] group exhibited the smallest increase in tumor volume compared to the other groups (Figure 4B). Additionally, the weight of excised tumors on day 15 was significantly lower in the tEx[miR] group, indicating the efficacy of this treatment (Figure 4C).

To further investigate the molecular effects of tEx[miR], the expression of tumor-related markers was examined in excised tumor tissues using real-time PCR. The tEx[miR] group showed the lowest expression of MACC-1, a marker associated with tumor progression and metastasis. Furthermore, this group also exhibited the lowest levels of the anti-apoptotic marker MCL-1 and the highest expression of the pro-apoptotic marker BAX, suggesting enhanced apoptotic activity (Figure 4D). Western blot analysis was performed on the excised tumor tissues to validate the PCR results and to assess the protein levels of key apoptotic markers. The tEx[miR] group displayed the lowest expression of MACC-1 and anti-apoptotic markers MCL-1 and Bcl-xL. Conversely, the pro-apoptotic marker BIM was significantly upregulated in the tEx[miR] group, corroborating the PCR findings and indicating a shift towards apoptosis in these tumors (Figure 4E).

### 2.5. Immunohistochemistry and Immunofluorescence Analyses in Mouse Colorectal Cancer Xenograft Models

The expression of key apoptotic markers was investigated in excised tumor tissues from each treatment group using immunohistochemistry and immunofluorescence analyses. Immunohistochemistry was employed to assess the expression of the anti-apoptotic marker Bcl-xL and the pro-apoptotic marker Bax. The immunohistochemistry results revealed that the tEx[miR] group exhibited the most significant reduction in Bcl-xL expression and the highest increase in Bax expression compared to the other groups (*p* < 0.05) (Figure 5A).

Similarly, immunofluorescence analysis was conducted to compare the expression of the anti-apoptotic marker MCL-1 and the pro-apoptotic marker Bax. The tEx[miR] group demonstrated the lowest expression of MCL-1 and the highest expression of Bax (*p* < 0.05) (Figure 5B). These findings indicate that treatment with tEx[miR] significantly enhances pro-apoptotic activity while reducing anti-apoptotic signals, demonstrating the highest anticancer effect in the mouse CRC xenograft model.

### 2.6. Analysis of Apoptosis in Mouse Tumor Tissues and miRNA Expression in HCT116 Cell Lines

Subsequently, apoptosis in tissues derived from each mouse group was assessed using the TUNEL assay (Figure 6A). The analysis revealed a significantly higher number of apoptotic cells in the tEx[miR] group compared to the control and Ex[miR] groups. Specifically, the TUNEL assay indicated a pronounced increase in apoptotic cell count in the tEx[miR]-treated group (*p* < 0.05), demonstrating the superior efficacy of tEx[miR] in inducing apoptosis. This enhancement in apoptosis highlights the potential of tEx[miR] as a more effective therapeutic strategy compared to Ex[miR].

To determine the impact of targeting MACC1 through tEx[miR], miRNA sequencing on CRC cell lines was conducted. In this method, miRNA expression was analyzed in HCT116 cell lines treated with Ex[miR]. The Venn diagram in Figure 6B illustrates the overlap of differentially expressed miRNAs among the three treatment groups in HCT116 cell lines: NC (negative control), Ex (HCT116 cells treated with control Ex), and Ex[miR] (HCT116 cells treated with Ex[miR]). It shows the unique and shared miRNAs among these groups. Specifically, 837 miRNAs were common across all three conditions, indicating a core set of miRNAs consistently expressed in HCT116 cells regardless of treatment. Unique miRNAs for each condition included 742 in the Ex group, 489 in the Ex[miR] group, and 588 in the NC group. Additionally, there were 88 miRNAs shared between the Ex and Ex[miR] groups, 66 shared between the Ex[miR] and NC groups, and 53 shared between the Ex and NC groups. These overlaps highlight the differential expression patterns induced by each treatment.

The miRNA expression profiles under different treatment conditions are depicted in a heatmap (Figure 6C). The heatmap was generated using two-way hierarchical clustering with Z-score normalization (log2 based) and highlights 15 mature miRNAs that satisfied the fold change (fc2) and raw *p*-value criteria. The three treatment groups are color-coded: NC (negative control) in green, Ex (HCT116 cells treated with control Ex) in red, and Ex[miR] (HCT116 cells treated with Ex[miR]) in blue. Key observations from the heatmap revealed distinct miRNA expression patterns among the groups. Specifically, the NC group exhibited baseline levels of miRNA expression without exosomal treatment. The Ex group showed varied miRNA expression, indicating that control exosomes can influence miRNA levels to a significant degree. The most significant changes were observed in the Ex[miR] group, which showed increased expression of specific miRNAs such as hsa-miR-143-5p and hsa-miR-143-3p. These miRNAs have been shown to enhance anticancer effects [19], suggesting the effective modulation of miRNA expression by miR-143-loaded tEx. These differential expression patterns highlight the therapeutic potential of miR-143-loaded exosomes.

## 3. Discussion

This research investigates the therapeutic delivery of MACC1-suppressing miRNA using TM4SF5-targeting Ex in CRC mouse models. The study focused on evaluating the targetability and efficacy of intravenously administered tEx[miR-143]. Both in vitro and in vivo assessments were performed using mouse CRC xenograft models. tEx demonstrated higher targetability compared to control (nontargeted) Ex and showed promising anti-tumor effects, as evidenced by the smallest tumor size, the lowest tumor growth rate, and the lowest weight of excised tumors in the tEx[miR-143] group. Furthermore, tEx[miR-143] treatment led to the lowest expression of MACC-1 and anti-apoptotic markers MCL-1 and Bcl-xL, while inducing the highest expression of pro-apoptotic markers BAX and BIM. These findings indicate the potential of tEx[miR-143] as a therapeutic strategy for CRC.

Targeting TM4SF5 as a surface marker on CRC cells holds considerable promise, given its potential contribution to diagnosis and treatment strategies. Studies have reported that TM4SF5 is highly expressed in up to 80% of CRC cells, making it a promising target for cancer-specific interventions [17]. Furthermore, TM4SF5 is also expressed in various other tumor types, such as liver (90%), gastric (75%), and pancreatic (70%) cancers, indicating its potential relevance across a wide range of cancers [18]. A crucial aspect to consider when targeting TM4SF5 is its expression in normal cells. Research has shown that TM4SF5 expression is relatively low or absent in healthy cells, with only about 5% of normal cells expressing TM4SF5 at detectable levels [20]. In summary, focusing on TM4SF5 as a surface marker for CRC cells has considerable potential due to its high expression in cancerous cells and limited presence in normal cells, offering a valuable strategy for targeted diagnosis and treatment.

Inhibiting the function of MACC-1 has emerged as an effective approach in the treatment of CRC. MACC-1 is a transcriptional regulator involved in various cancer-related processes, including cell proliferation, migration, and invasion (Stein et al., 2009) [21]. It is predominantly expressed in tumor cells, with high expression levels observed in CRC [22]. Studies have reported that increased MACC-1 expression correlates with poor prognosis and metastasis in CRC patients [23]. In contrast, its expression in normal cells is significantly lower, suggesting a potential therapeutic target for cancer treatment [24]. MicroRNAs that inhibit MACC-1 function have garnered attention as potential therapeutic agents. For instance, miR-143 and miR-145 have been shown to target and suppress MACC-1 expression, effectively inhibiting CRC cell proliferation, migration, and invasion [25]. By incorporating these microRNAs into targeted therapeutic strategies, it is anticipated that they will significantly improve CRC treatment outcomes.

It is postulated that tEx[miR-143] exhibits its anticancer effect by binding to TM4SF5 on the surface of CRC cells, delivering miR-143, which suppresses the function of MACC1. This, in turn, inhibits the EMT progression, ultimately leading to a reduction in cancer cell growth and metastasis (Figure 7). The mechanism by which MACC1 increases EMT can be divided into two pathways: (1) direct interaction with EMT markers, and (2) indirect regulation through Twist1/2. In the direct interaction pathway, MACC1 modulates the expression of EMT markers, such as E-cadherin, N-cadherin, and vimentin, by directly binding to their respective promoters or interacting with transcription factors that regulate their expression [26,27,28]. This leads to the suppression of epithelial markers and the induction of mesenchymal markers, thereby promoting EMT. In the indirect regulation pathway, MACC1 acts through Twist1/2, a pair of transcription factors known to play crucial roles in EMT. MACC1 upregulates the expression of Twist1/2, which, in turn, represses the expression of epithelial markers, such as E-cadherin, and promotes the expression of mesenchymal markers, such as N-cadherin and vimentin [29,30]. This process facilitates the EMT and contributes to cancer progression and metastasis.

We believe that this study offers meaningful insights into CRC research and treatment, as well as the broader field of cancer therapy. In this study, a novel therapeutic strategy was devised to target TM4SF5-expressing cancer cells and suppress MACC-1 function using microRNA-encapsulated Ex, demonstrating promising anti-tumor effects both in vitro and in vivo. TM4SF5 is highly expressed in many tumors and scarcely found in normal cells, indicating the potential applicability of this approach to a variety of cancer types beyond CRC [29,31]. Additionally, our strategy serves as a versatile platform technology, capable of accommodating different cargoes, such as miR-143 or other therapeutic molecules, for targeted delivery to cancer cells. The development of this targeted therapy not only has the potential to improve the efficacy of CRC treatment but also to reduce unwanted side effects on normal cells, thus enhancing patient outcomes.

One limitation of our study was the lack of dramatic tumor size reduction observed in the group treated with tEx[miR-143]. This raises the question of whether targeted gene therapy is less effective than targeted drug therapy for cancer treatment [32,33,34]. However, it is important to note that measuring efficacy solely by tumor size reduction may not fully capture the potential benefits of a therapy. For example, in the case of immune checkpoint inhibitors, the primary endpoint for evaluating efficacy is often overall survival rather than tumor size [35,36,37]. Whereas targeted drug therapy typically works by directly inhibiting a specific protein, leading to a more immediate therapeutic effect, targeted gene therapy involves modulating the expression of a specific gene to regulate the amount of protein produced, potentially resulting in longer-lasting therapeutic effects [32,33]. The effectiveness and safety of targeted gene therapy for tumor diseases require further research to be better understood.

In conclusion, our study demonstrates the potential of TM4SF5-targeting Ex loaded with MACC1-suppressing miRNA as an innovative therapeutic approach for CRC. Our findings suggest that TM4SF5-targeting Ex[miR-143] have higher targetability and exhibit promising anti-tumor effects, as evidenced by significant reductions in tumor size, growth rate, and weight. Moreover, the treatment led to the downregulation of MACC-1 and anti-apoptotic markers while upregulating pro-apoptotic markers. These results provide a foundation for the development of new and effective therapies for CRC, which could improve patient outcomes and reduce unwanted side effects.

## 4. Materials and Methods

### 4.1. Cell Culture

Human adipose-derived stem cells (ASCs) were obtained from Hurim BioCell Co. (Seoul, Republic of Korea). ASCs were cultured in DMEM/low glucose (GibcoBRL, Waltham, MT, USA) supplemented with antibiotics (penicillin-streptomycin; GibcoBRL) at 37 °C in a humidified atmosphere with 5% CO_2_ in an incubator (Panasonic MCO-170AIC-PIC incubator; Panasonic Healthcare, Wood Dale, IL, USA). HCT116, HT29, and SW480 cells were procured from the Korea Cell Line Bank (KCLB, Seoul, Republic of Korea), and the Short Tandem Repeat (STR) identification data for these cell lines are provided in Appendix A. HCT116 cells were maintained in RPMI1640 (Hyclone, UT, USA) with 10% fetal bovine serum (Hyclone, Logan, UT, USA) and 1% penicillin-streptomycin (GibcoBRL) at 37 °C in a humidified atmosphere with 5% CO_2_ in an incubator.

### 4.2. Establishment of Doxorubicin-Resistant CRC Cells

To establish doxorubicin (Sigma-Aldrich, St. Louis, MO, USA)-resistant cell lines, HCT116 and HT29 cells were cultured under laboratory conditions. The cells were exposed to a stepwise increase in doxorubicin concentration, starting at 0.5 µM and gradually escalating from 5 µM to 8 µM, over a six-month period. The resulting doxorubicin-resistant CRC cells were then maintained by culturing them with a constant exposure to a lower dose (1 µM) of doxorubicin to sustain their resistance.

### 4.3. Production of TM4SF5-Targeted Ex (tEx)

Conditioned medium was generated from 90% confluent ASC cultures that had been subjected to starvation conditions (absence of FBS) for 24 h after transfecting 4 µg of pDisplay-TM (TM4SF5 targeted peptide) into ASCs for 24 h. The conditioned medium was collected and centrifuged at 2500× *g* for 15 min at 4 °C to eliminate cell debris. Exosome isolation reagent was added to the conditioned medium, and the solution was incubated overnight at 4 °C. Exosome isolation was conducted from the conditioned medium using differential centrifugation at 10,000× *g* for 60 min at 4 °C. The precipitated exosomes were recovered by standard centrifugation at 10,000× *g* for 60 min, and the pellet was resuspended in PBS. Utilizing the ExofectTM kit, miR-143 (Bioneer, Daejeon, Republic of Korea) was encapsulated into Ex or tEx, resulting in the formation of Ex[miR-143] or tEx[miR-143], respectively.

### 4.4. Exosome Characterization

Nanoparticle tracking analysis (NTA) was performed using a Zetaview instrument (Particle Metrix GmbH, Ammersee, Bavaria, Germany). For each measurement, 11 videos lasting 1 cycle were captured. All measurements conformed to the quality criteria of 50–150 particles/frame, a concentration of 10^7^ particles/mL, and valid tracks > 20%. Videos were analyzed by the built-in Zeta-view software (Version 8.05.16 SP3) upon capture.

### 4.5. Transmission Electron Microscopy (TEM)

For TEM analysis, PBS-resuspended Ex were adsorbed onto Formvar carbon-coated grids for 10 min, and excess liquid was removed using filter paper. A 1% uranyl acetate solution was employed as a negative stain for 10 min, and excess liquid was eliminated using filter paper. The grid was allowed to dry at room temperature. The adsorbed Ex were examined with a JEM1010 transmission electron microscope (JEOL Ltd., Tokyo, Japan) at 60 kV.

### 4.6. Real-Time PCR

Total RNA was extracted from HCT116 cells and mouse liver tissues using TRIzol reagent (Invitrogen, Carlsbad, CA, USA). Reverse transcription was performed with 1 µg RNA using an RT-premix kit (TOYOBO, Osaka, Japan) following the manufacturer’s instructions. SYBR Green real-time quantitative polymerase chain reaction (PCR) was conducted using the following primers: human Mcl-1 forward 5′-GGGCAGGATTGTGACTCTCATT-3′ and reverse 5′-GATGCAGCTTTCTTGG TTTATGG-3′; human Bax forward 5′-TGGAGCTGCAGAGGATGATTG-3′ and reverse 5′-GAAGTTGCCGTCAGAAAACATG-3′; human GAPDH forward 5′-GCACCGTCAAGGCTGAGAAC-3′ and reverse 5′-TGGTGAAGACGCCAGTGGA-3′; mouse Mcl-1 forward 5′-AAAGGCGGCTGCATAAGTC-3′ and reverse 5′-TGGCGGTATAGGTCGTCCTC-3′; mouse Bax forward 5′-CTGCAGAGGATGATTGCCG-3′ and reverse 5′-TGCCACTCGGAAAAAGACCT-3′; mouse GAPDH forward 5′-CGACTTCAACAGCAACTCCCACTCTTCC-3′ and reverse 5′-TGGGTGGTCCAGGGTTTCTTACTCCTT-3′. Reactions were performed using the Applied Biosystems StepOnePlus Real-Time PCR System (Thermo, Carlsbad, CA, USA).

### 4.7. Wound Healing Assay

HCT116 colon cancer cells were grown to confluence in 48-well plates, and the medium was replaced with serum-free media. The cells were incubated for an additional 48 h. Cell monolayers were wounded and treated with test agents. A defined area of the wound was photographed under phase-contrast microscopy before treatment. The predetermined wound area was re-photographed after 48 h, and the remaining wound area was determined by image analysis. Wound closure rate was calculated as [(initial − final)/initial] × 100.

### 4.8. Western Blot Analysis

HCT116 colon cancer cells and mouse tissues were lysed using the EzRIPA Lysis kit (ATTO Corporation; Tokyo, Japan) and quantified by Bradford reagent (Bio-Rad, Hercules, CA, USA). Proteins were visualized by western analysis using primary antibodies (1:1000 dilution) from Cell Signaling Technology (Beverly, MA, USA), followed by HRP-conjugated secondary antibodies (1:2000 dilution) from Vector Laboratories (Burlingame, CA, USA). Specific immune complexes were detected using the Western Blotting Plus Chemiluminescence Reagent (Millipore, Bedford, MA, USA). Primary antibodies used were MACC1 (Mybiosource, San Diego, CA, USA; #MBS9611885), MCL-1 (Cell signaling Technology, Danvers, MT, USA; #5453), BCL-Xl (Cell signaling; #2764p), PUMA (Cell signaling; #12450), BIM (Cell signaling; #2933s), and β-actin (Sigma-Aldrich; #A5541). Gel images were captured using the Bio-Rad ChemiDoc MP Imaging System (Bio-Rad Laboratories, Hercules, CA, USA).

### 4.9. Immunofluorescence and Immunohistochemical Analysis

For immunofluorescence and immunohistochemical analysis, formalin-fixed, paraffin-embedded tissue sections were deparaffinized, rehydrated in an ethanol series, and subjected to epitope retrieval using standard procedures. Antibodies to MCL-1 (Santa Cruz Biotechnology, Dallas, TX, USA; sc-377487), BAX (Santa Cruz; sc-7480), E-cadherin (Santa Cruz; sc-8426), Snail (GeneTex, Irvine, CA, USA; gtx125918) and GAPDH (Santa Cruz; sc-365062) were used for immunofluorescence staining, and antibodies to MACC1 (Mybiosource; MBS9611885), Bcl-xL (Cell signaling Technology, Danvers, MT, USA; 2764p), and BAX (Santa Cruz; sc-7480) were used for immunohistochemical staining. Samples were then examined under a laser-scanning microscope (Eclipse TE300; Nikon, Tokyo, Japan) to analyze the expression of these antibodies.

### 4.10. Cell Viability Assay

HCT116, HT29, and SW480 cell viability were assessed using the Ez-cytox Cell Viability Assay Kit (Itsbio, Seoul, South Korea) following the manufacturer’s protocol.

### 4.11. Flow Cytometry

The proportion of target exosomes in the produced exosomes was confirmed using flow cytometry. Targeted exosomes produced by asc cells transfected with pDisplay-TM4SF5_P_ were stained with anti-myc antibody (R&D systems, Minneapolis, MN, USA). Exosomes produced by human ASC cells were stained with anti-CD63 antibodies (BD Pharmingen: San Francisco, CA, USA) and anti-CD81 antibodies (BD Pharmingen) to confirm the exosome marker. After incubation for 60 min in the dark at 5 °C, the cells were analyzed using Attune xT acoustic focusing cytometer (Thermo Fisher Scientific, Waltham, MA, USA).

### 4.12. RNA Sequencing Library Construction and Validation

Following the manufacturer’s, the RNA was subjected to a sequencing library construction using the NEBNext Multiplex Small RNA Library kit for Illumina (Sandiego, CA, USA). Briefly, the 3′ adapter was ligated to the 3′ ends of all miRNAs, and the RT primer was bound to a region of the 3′ adapter for cDNA synthesis. The 5′ adapter was ligated to the 5′ end of mature miRNAs. RT reaction was used to create single-stranded cDNA. The products were then purified and enriched with PCR to create the final cDNA library. The libraries were gel purified by BluePippn (Sage Science, Beverly, MT, USA) and validated by checking the size, purity, and concentration on the Agilent BioAnalyzer (Santa Clara, CA, USA). The libraries were quantified using KAPA Library Quantification kits for Illumina Sequencing (Roche Sequencing Solutions, Pleasanton, CA, USA) platforms according to the qPCR Quantification Protocol Guide. Indexed libraries were pooled in equimolar amounts, and sequence on an Illumina NovaSeq instrument.

### 4.13. Spheroid Formation and Viability Assessment

HCT116 cells (1 × 10^3^) were seeded to form spheroids. After 24 h, the spheroids were treated with exosomes loaded with miR-143. Cell viability was assessed using a LIVE/DEAD cytotoxicity assay (Invitrogen) following a 30 min incubation at 37 °C. Live cells with intact membranes cleave the non-fluorescent Calcein-AM into a brightly fluorescent Calcein product. Fluorescence intensity was measured using a Cell Voyage CQ1 microscope (Yokogawa Electron Co., Musashino, Japan) to quantify organoid viability.

### 4.14. Animal Study Design

Five-week-old male BALB/c nude mice (Orient Bio, Seongnam, Republic of Korea) were used for comparative modeling of subcutaneous tumor growth. HCT116 cells (5 × 10^6^) were subcutaneously injected into each mouse. Animal studies were conducted in compliance with the guidelines of the Institute for Laboratory Animal Research, the Catholic University of Korea (IRB No: CUMC-2022-0077-02). The mice were weighed twice a week. Seven days after tumor cell injection, all mice had measurable tumors. In vivo targetability was assessed by randomly grouping mice (n = 3 per group) and treating them intraperitoneally with PBS (control), Ex (2 × 10^6^ Ex particles in 100 µL PBS, once), tEx (2 × 10^6^ Ex particles in 100 µL PBS, once), Ex (miR-143) [2 × 10^6^ Ex particles in 100 µL PBS, once], or tEx (miR-143) [2 × 10^6^ Ex particles in 100 µL PBS, once] for 1 day. Human ASC-derived Ex were labeled with the ExoGlowTM Ex Labeling Kit (SBI Biosciences, CA, USA), and unbound dye was removed via ultracentrifugation and washing. ExoGlow-Vivo-labeled Ex and tEx showed robust signal in vivo. Animals were imaged at various time points using an IVIS In vivo Imaging System (PerkinElmer, Waltham, MA, USA). In vivo efficacy was evaluated by randomly grouping mice (n = 5 per group) and treating them intraperitoneally with PBS (control), Ex (2 × 10^6^ Ex particles in 100 µL PBS, twice a week), tEx (2 × 10^6^ Ex particles in 100 µL PBS, twice a week), Ex (miR-143) [2 × 10^6^ Ex particles in 100 µL PBS, twice a week], or tEx (miR-143) [2 × 10^6^ Ex particles in 100 µL PBS, twice a week] for 21 days. Ex and tEx were labeled using the ExoGlowTM Ex Labeling Kit (SBI Biosciences, CA, USA). ExoGlow-Vivo-labeled Ex and tEx showed robust signals in vivo. The tumor size was measured twice a week using a caliper, and tumor volume (V) was calculated using the formula: length × width^2^ × 0.5236. After completion of treatment, all mice were euthanized.

### 4.15. Statistical Analysis

All data were analyzed using SPSS 11.0 software (SPSS Inc., Chicago, IL, USA) and presented as mean ± standard deviation (SD). Statistical comparisons among groups were determined using the Kruskal–Wallis test. Probability values of *p* < 0.05 were considered statistically significant.

## Figures and Tables

**Figure 1 ijms-25-09232-f001:**
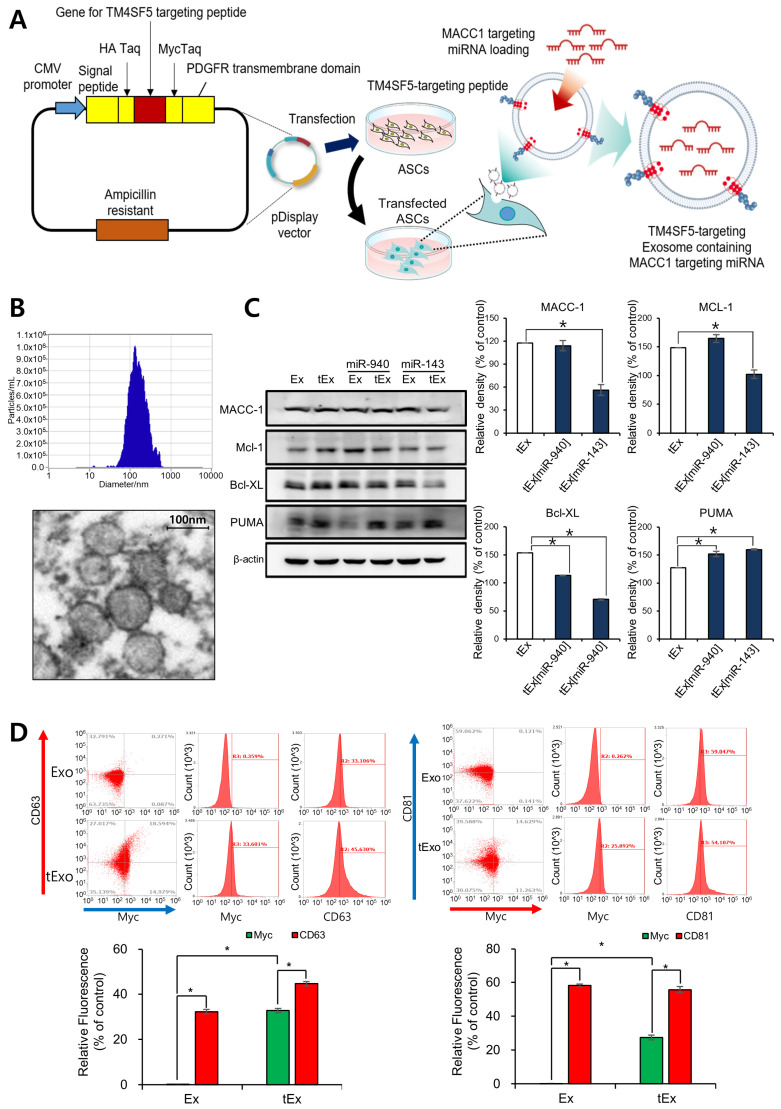
Characterization and functional evaluation of TM4SF5-targeting Ex encapsulating MACC1-suppressing miRNAs in HCT116 cells. (**A**) Designing and development of TM4SF5-targeting Ex (MACC1-suppressing miRNA). First, the DNA sequence encoding the TM4SF5-targeting peptide was inserted into the pDisplay vector, which contains a portion of the PDGFR transmembrane domain, using restriction enzymes. This recombinant vector was then used to transfect adipose-derived stem cells (ASCs). The transfected ASCs expressed the TM4SF5-targeting peptide on their cell membrane, and the Ex released from these cells displayed the TM4SF5-targeting peptide on their surface. Finally, MACC1-suppressing miRNA was loaded into the Ex to complete the fabrication of TM4SF5-targeting exosomes containing MACC1-suppressing miRNA. (**B**) Size distribution of nanoparticles as determined by Zetaview analysis, showing a mean size of 148 ± 84 nm, consistent with the expected size range for exosomes (top) and representative TEM image of the isolated nanoparticles (bottom), Scale bar: 100 nm. (**C**) Western blot analysis comparing the therapeutic effects of TM4SF5-targeted Ex and control Ex loaded with miR-940 and miR-143 in HCT116 cells. While no significant difference in MACC1 expression inhibition was observed between TM4SF5-targeted Ex (miR-940) and TM4SF5-targeted Ex (miR-143), the latter showed a greater reduction in anti-apoptotic markers MCL-1 and Bcl-xL, and increased pro-apoptotic marker PUMA expression (*Ps* < 0.05). Relative densities of individual markers had been quantified using ImageJ software (version 1.54d) and then were normalized to that of β-actin in each group. (**D**) Flow cytometry analysis of exosome markers (CD63, CD81) and the TM4SF5-targeted exosome marker (myc) in Ex and tEx groups. CD63 and CD81 expression levels were similar between Ex and tEx groups. However, myc expression was significantly higher in tEx compared to Ex in both CD63-positive and CD81-positive cells (*p* < 0.05). Values are presented as mean ± standard deviation of three independent experiments. * *p* < 0.05.

**Figure 2 ijms-25-09232-f002:**
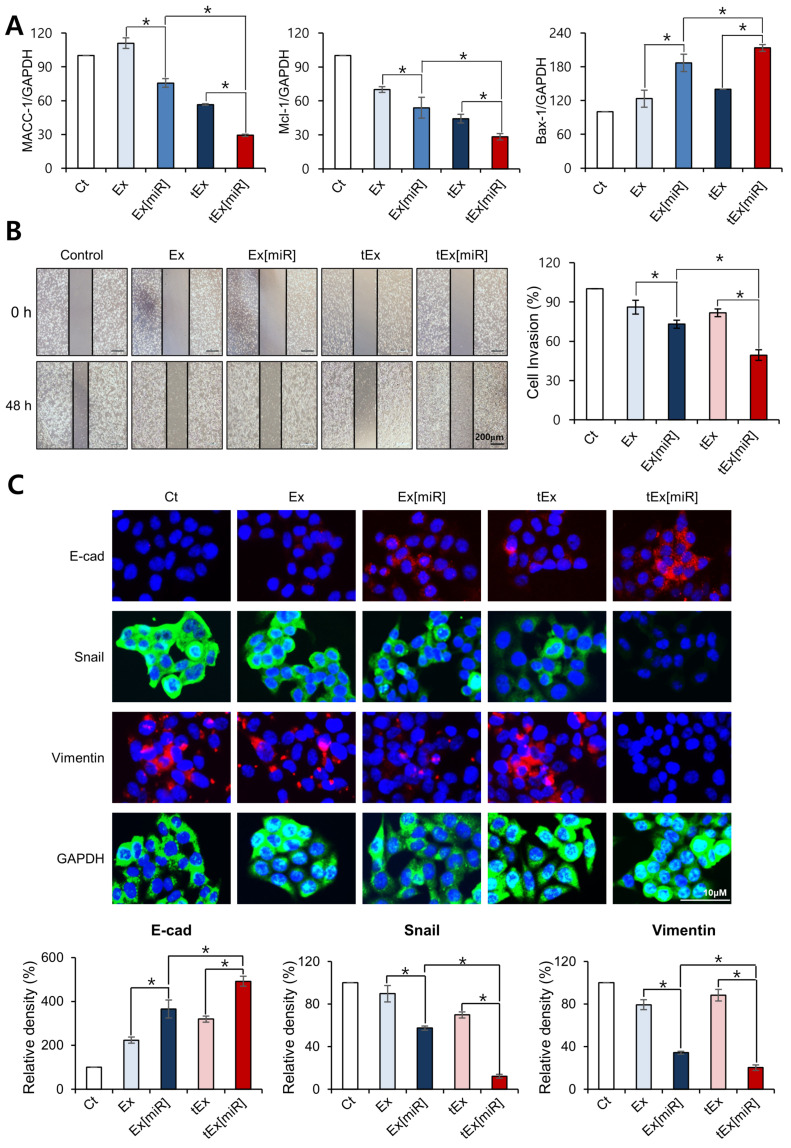
In vitro assessment of TM4SF5-targeted exosomes with miR-143 on cancer cell behavior. (**A**) Real-time PCR analysis comparing four groups: Ex, Ex[miR], tEx, and tEx[miR]. tEx[miR] exhibited the most significant decrease in MACC-1 expression, the lowest anti-apoptotic marker MCL-1 expression, and the highest pro-apoptotic marker BAX-1 expression (*Ps* < 0.05). (**B**) Wound healing assay evaluating the effects of each treatment on HCT116 cell migration and invasion. Representative images [left] were taken at 0 h and 48 h for each treatment group: control (Ct), Ex, Ex[miR-143], tEx, and tEx[miR-143]. The images show the extent of wound closure over the 48 h period. Right panel shows the quantification of cell invasion as a percentage of the initial wound area. The tEx[miR-143] group exhibited the most significant inhibition of cell migration and invasion compared to other groups, with a statistically significant reduction in cell invasion percentage (*p* < 0.05). (**C**) Immunofluorescence staining of HCT116 cells for E-cadherin, Snail, Vimentin, and GAPDH following treatments. E-cadherin (red), Snail (green), and Vimentin (red) expressions are shown, with nuclei stained blue with DAPI. The relative density of each marker is quantified in the bar graphs below the images. The tEx[miR] group exhibits increased E-cadherin and decreased Snail and Vimentin levels, indicating a significant inhibition of EMT. Values are presented as mean ± standard deviation of three independent experiments. * *p* < 0.05.

**Figure 3 ijms-25-09232-f003:**
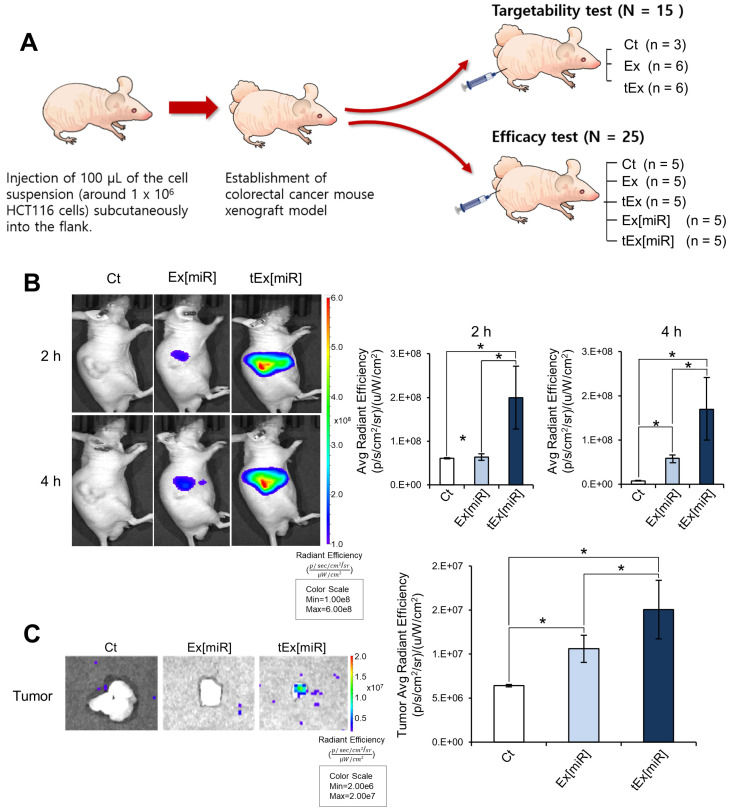
In vivo assessment of targetability for tEx[miR] in mouse colorectal cancer xenograft models. (**A**) Schematic representation of the experimental design for independent evaluation of targetability and efficacy. Colorectal cancer mouse xenograft models were established by subcutaneous injection of 1 × 10^6^ HCT116 cells into the flank of each mouse. The mice were then divided into two major groups: one group (N = 15) was utilized to evaluate targetability, and the other group (N = 25) was employed to assess efficacy. In the targetability test group, mice were further divided into Ct (n = 3), Ex (n = 6), and tEx (n = 6). In the efficacy test group, mice were divided into Ct (n = 5), Ex (n = 5), tEx (n = 5), Ex[miR] (n = 5), and tEx[miR] (n = 5). (**B**) Fluorescently labeled Ex were imaged using the IVIS Imaging system, revealing higher targetability of tEx[miR] compared to the control (Ct) and Ex[miR] groups. Images were taken at 2 h and 4 h post injection. The left panel shows representative images of the mice at these time points. The right panel displays the quantification of average radiant efficiency (p/s/cm^2^/sr)/(μW/cm^2^) in the whole body, revealing significantly higher targetability of tEx[miR] compared to Ex[miR] at both 2 h and 4 h. (**C**) Comparison of radiant efficiency in excised tumors from xenograft models, demonstrating higher targetability of tEx[miR] compared to Ct and Ex[miR] groups. Excised tumors from the xenograft models were also imaged using the IVIS Imaging System. The left panel shows representative images of the tumors, indicating higher accumulation of tEx[miR] compared to Ex[miR]. The right panel quantifies the average radiant efficiency in the tumors, further confirming the enhanced targetability of tEx[miR]. Values are presented as mean ± standard deviation of three independent experiments. * *p* < 0.05.

**Figure 4 ijms-25-09232-f004:**
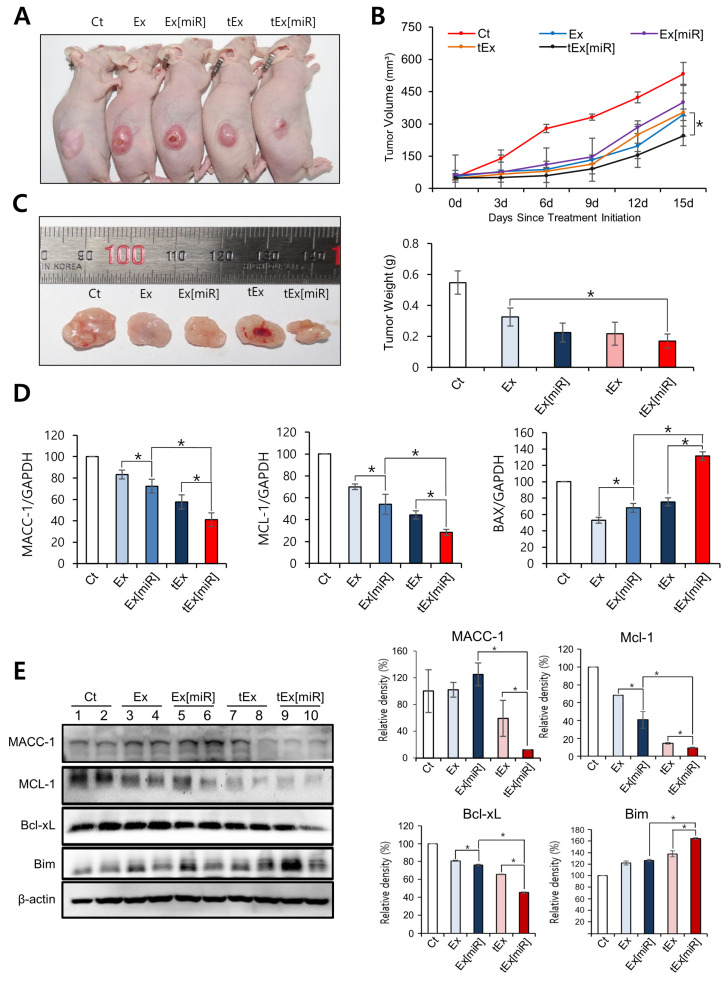
Evaluation of in vivo efficacy of TM4SF5-targeting Ex (miR-143) in mouse colorectal cancer xenograft models. (**A**) Representative images of mouse colorectal cancer xenograft models treated with different groups: Ct, Ex, Ex[miR], tEx, and tEx[miR], showing the smallest tumor sizes in the tEx[miR] group on day 15 post treatment. (**B**) Graph showing the changes in tumor volume over 15 days of treatment. The tEx[miR] group exhibited the smallest increase in tumor volume compared to other groups. (**C**) Comparison of excised tumor weights from the different treatment groups on day 15. The tEx[miR] group had significantly lower tumor weights. Representative images of the excised tumors are shown on the left, with quantification on the right. (**D**) Quantification of MACC-1, MCL-1, and BAX expression levels in excised tumor tissues from different treatment groups using real-time PCR. The tEx[miR] group showed the lowest MACC-1 and MCL-1 expression and the highest BAX expression. (**E**) Western blot images and quantification of MACC-1, MCL-1, Bcl-xL, and BIM protein levels in excised tumor tissues. The tEx[miR] group demonstrated the lowest MACC-1, MCL-1, and Bcl-xL levels, and the highest BIM levels, indicating enhanced apoptotic activity. Values are presented as mean ± standard deviation of three independent experiments. * *p* < 0.05.

**Figure 5 ijms-25-09232-f005:**
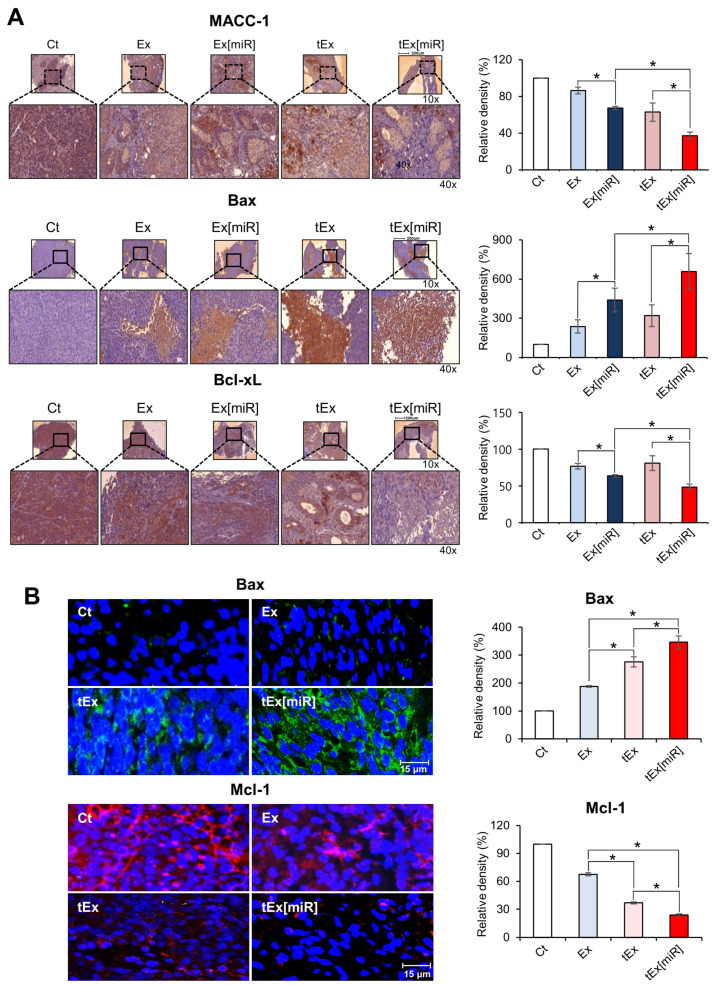
Immunohistochemistry and immunofluorescence analyses of anti-apoptotic and pro- apoptotic marker expression in mouse colorectal cancer xenograft model. (**A**) Immunohistochemical staining of excised tumor tissues for MACC-1, Bax, and Bcl-xL expression in different treatment groups: Ct, Ex, Ex[miR], tEx, and tEx[miR]. The tEx[miR] group showed the greatest reduction in Bcl-xL expression and the highest increase in Bax expression. Representative images are shown at 10× and 40× magnification. The bar graphs on the right quantify the relative density of marker expression. Values are presented as mean ± standard deviation. * *p* < 0.05. (**B**) Immunofluorescence staining of excised tumor tissues for Bax and MCL-1 expression in different treatment groups. The tEx[miR] group exhibited the lowest MCL-1 expression and the highest Bax expression. Representative images are shown at the 15 µm scale. The bar graphs on the right quantify the relative density of marker expression. Values are presented as mean ± standard deviation of three independent experiments. Percentages of immunoreactive areas were measured using NIH ImageJ and expressed as relative values to those in normal livers. * *p* < 0.05.

**Figure 6 ijms-25-09232-f006:**
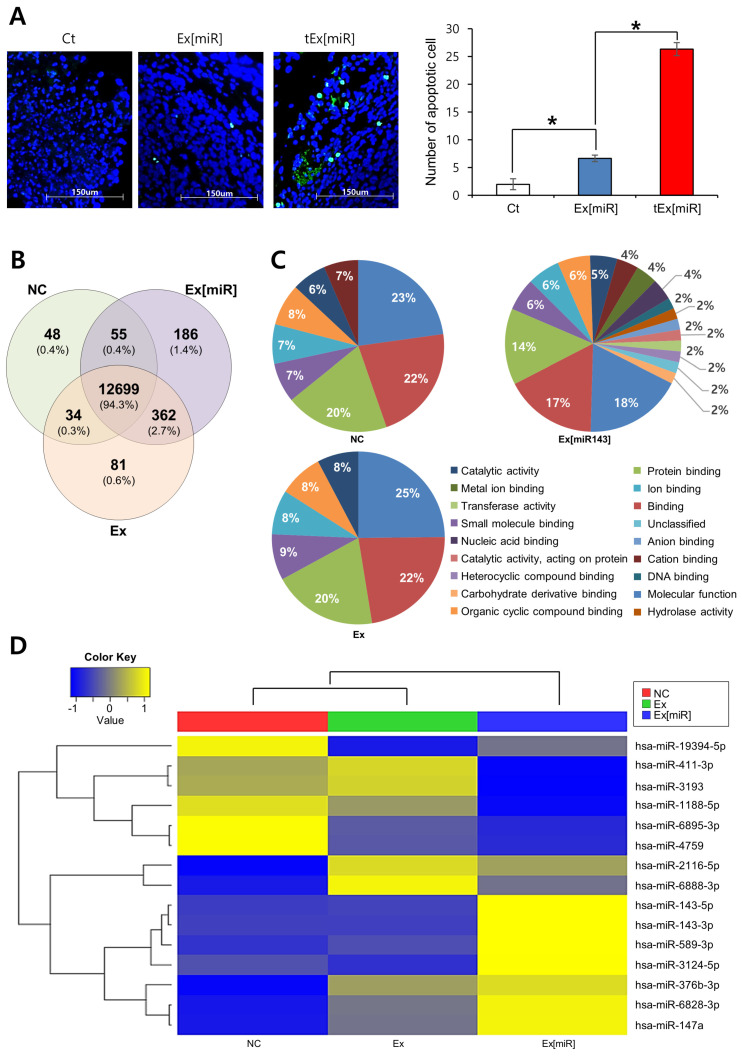
Analysis of apoptosis in mouse tumor tissues and miRNA expression in HCT116 cell lines. (**A**) TUNEL assay demonstrating enhanced apoptosis in the tEx[miR]-treated group. Representative immunofluorescence images of the TUNEL assay in tissue sections from the control (Ct)-, Ex[miR]-, and tEx[miR]-treated groups [Left]. Apoptotic cells were stained green, and nuclei were stained blue with DAPI. Scale bar: 100 μm. Quantification of apoptotic cells in tissue sections [Right]. The number of apoptotic cells was significantly higher in the tEx[miR] group compared to the Ct and Ex[miR] groups. Values are presented as mean ± standard deviation of three independent experiments. * *p* < 0.05. (**B**,**C**) Venn diagram of differentially expressed miRNAs in HCT116 cell line. The Venn diagram shows the overlap of differentially expressed miRNAs among three treatment groups in HCT116 cell lines: NC (negative control), Ex (HCT116 cells treated with Ex[miR]), and tEx (HCT116 cells treated with tEx[miR]). Each circle represents a treatment group, with the numbers indicating the count of unique and shared miRNAs among the groups. (**D**) The heatmap illustrates the miRNA expression profiles in HCT116 cell lines under different treatment conditions, including NC (red), Ex (green), and tEx (blue). The analysis uses two-way hierarchical clustering with Z-score normalization (log2 based) and highlights 15 mature miRNAs that met the fold change (fc2) and raw *p*-value criteria. The color scale represents the Z-score values, with blue indicating lower expression and yellow indicating higher expression.

**Figure 7 ijms-25-09232-f007:**
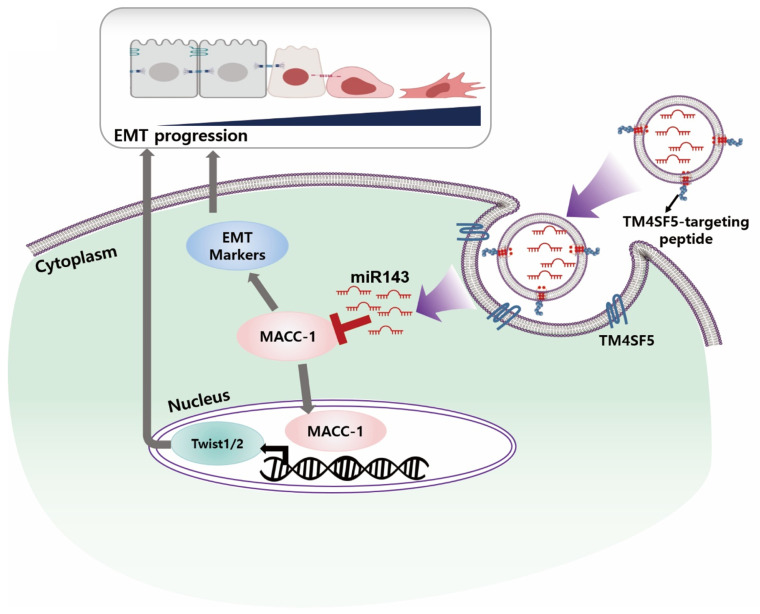
Proposed mechanism of TM4SF5-targeting Ex (miR-143) suppressing the EMT in the colorectal cancer. MACC1 enhances EMT through two pathways: (1) direct interaction with EMT markers, modulating their expression, and (2) indirect regulation via upregulation of Twist1/2 transcription factors, which further alter the expression of epithelial and mesenchymal markers. TM4SF5-targeting Ex (miR-143) exhibits its anticancer effect by binding to TM4SF5 on the surface of colorectal cancer cells, delivering miR-143, which suppresses the function of MACC1. This, in turn, inhibits the EMT progression, ultimately leading to a reduction in cancer cell growth and metastasis.

## Data Availability

The datasets generated and/or analyzed during the current study are available from the corresponding author upon reasonable request.

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
