# Peer review of "Innovative Therapeutic Delivery of Metastasis-Associated in Colon Cancer 1-Suppressing miRNA Using High Transmembrane 4 L6 Family Member 5-Targeting Exosomes in Colorectal Cancer Mouse Models"

_ijms, 2024, doi:10.3390/ijms25179232_

Round 1
Reviewer 1 Report
Comments and Suggestions for Authors
This study investigates the therapeutic targeting of MACC1-silencing miRNA using TM4SF5-targeting Ex (tEx[miR-143]) in CRC cell lines and CRC mouse models. In the targeting experiment, tEx demonstrated high efficacy. The tEx[miR-143] treatment resulted in the lowest expression of MACC-1 and the anti-apoptotic MCL-1 and Bcl-xL markers, while it induced the highest expression of the pro-apoptotic BAX and BIM markers. The results indicate the potential of tEx[miR-143] as a therapeutic strategy for CRC.
The objectives, experimental protocol, experimental design, and methodologies are all state-of-the-art and adequate methods, and the statistical calculations used are correct.
Images and figures are illustrative. The original WB images are also correct.
The results of this research article are very important; they encourage further targeted miRNA-containing exosome experiments, not only in CRC but in other tumor diseases. Furthermore, these results could potentially serve as the foundation for future clinical trials.
However, some corrections and additions are necessary before acceptance for publication.
The article tangentially mentions the use of HT29 and SW480 cells. The authors described that the results obtained with these cells were trend-wise similar to those observed with HCT116 cells. Why were only trend-like differences observed? What could be the reason for these results not being statistically significant?
Why were only HCT116 cells used in the mouse model experiments? Why aren't the other two cell lines?
There are several references to Supplementary Figures in the article; however, I was only able to download the article and the original Western blot images. Where are the Supplementary figures? Uploading them is essential.
In the legend of Figure 5, TUNNEL must be changed to TUNEL.
The above corrections need to be made.
Reviewer 2 Report
Comments and Suggestions for Authors
In their manuscript, Choi and colleagues investigated a novel approach to treat colorectal cancer in vitro and in mouse models.
They trasfected adipose-derived stem cells (ASCs) to expressed the TM4SF5-targeting peptide on their cell membrane, and the released exosome (Ex) displayed the TM4SF5-targeting peptide on their surface.
MACC1-suppressing miRNA was encapsulated within the Ex to complete the fabrication of tEx[MACC1-suppressing miRNA, miR-143] (tEx[miR-143]).
The anticancer effects of these Ex were investigated in vitro using the human colorectal cell line HCT116 and in vivo using colorectal cancer mouse xenograft models.
Western blot analysis comparing the effects of TM4SF5-targeted Ex and control Ex loaded with miR-940 and miR-143 in HCT116 cells showed a reduction in anti-apoptotic markers MCL-1 and Bcl-xL, and increased pro-apoptotic marker PUMA expression.
In the in vivo assessment, administration of TM4SF5-targeting 29 Ex[miR-143], referred to as tEx[miR-143], resulted in the smallest tumor size, the lowest tumor growth rate, and the lightest excised tumors compared to other treatments.
The therapeutic approach based on the use of engineered vesicles is interesting and the results also appear to be partly interesting. Despite these premises, the manuscript cannot be accepted in this version.
Major points:
Although the authors report literature data where MACC is described to be a protein involved in the pathogenesis of CRC, as it has been associated with aggressive cancer cell behavior, enhanced metastatic potential, and reduced overall survival in patients and TM4SF5 is described as protein highly expressed protein in various solid tumors, including CRC, where it has been reported to be 50 overexpressed in 58% of cases the link between the two proteins is unclear. The two proteins to be targeted seem to be chosen at random, or it would be interesting to understand whether at the molecular level in vitro there is a common regulatory mechanism or whether there is some type of correlation at the expression level in CRC patients.
Fig 1 C. Bar chart displaying densitometry is not corresponding to the intensity band displayed in the image. Authors are asked to show an image of the western blotting in which the differences are more evident and in which the bands are more defined (perhaps increasing the quantity of lysate analysed could help). Moreover, the error bars on the histograms look too similar to each other
Fig 1D the error bars on the histograms look too similar to each other. Is this the calculated error of three biological experiments or the error of technical triplicate?
Fig 4 D, E: Why is the Bax transcript analyzed in panel D and the BIM protein analyzed in panel E?
Round 2
Reviewer 1 Report
Comments and Suggestions for Authors
The authors have replied to my questions correctly. The revised version of the manuscript is now acceptable for publication.